# Impact of Urinary Incontinence on Physical Function and Respiratory Muscle Strength in Incontinent Women: A Comparative Study between Urinary Incontinent and Apparently Healthy Women

**DOI:** 10.3390/jcm11247344

**Published:** 2022-12-10

**Authors:** Sirine Abidi, Amine Ghram, Sameh Ghroubi, Said Ahmaidi, Mohamed Habib Elleuch, Olivier Girard, Theodoros Papasavvas, Jari Laukkanen, Helmi Ben Saad, Beat Knechtle, Katja Weiss, Mehdi Chlif

**Affiliations:** 1Research Laboratory Education, Motricity, Sport and Health LR19JS01, High Institute of Sport and Physical Education of Sfax, University of Sfax, Sfax 3000, Tunisia; 2Research Laboratory “Heart Failure, LR12SP09”, Hospital Farhat HACHED of Sousse, Sousse 4031, Tunisia; 3Department of Pneumology, Principal Military Hospital of Instruction of Tunis, Tunis 1008, Tunisia; 4Healthy Living for Pandemic Event Protection (Hl-Pivot) Network, Chicago, IL 60612, USA; 5Research Laboratory: Evaluation and Management of Musculoskeletal System Pathologies, LR20ES09, Faculty of Medicine, University of Sfax, Sfax 3029, Tunisia; 6EA 3300, Exercise Physiology and Rehabilitation Laboratory, Sport Sciences Department, Picardie Jules Verne University, F-80025 Amiens, France; 7School of Human Sciences (Exercise and Sport Science), University of Western Australia, Perth, WA 6009, Australia; 8Department of Cardiac Rehabilitation, Heart Hospital, Hamad Medical Corporation, Doha 3050, Qatar; 9Faculty of Sport and Health Sciences, University of Jyväskylä, FI-40014 Jyväskylä, Finland; 10Department of Internal Medicine, Central Finland Healthcare District, FI-40530 Jyväskylä, Finland; 11Institute of Clinical Medicine, University of Eastern Finland, FI-70029 Kuopio, Finland; 12Laboratory of Physiology, Faculty of Medicine of Sousse, University of Sousse, Sousse 4002, Tunisia; 13Medbase St. Gallen Am Vadianplatz, 9100 St. Gallen, Switzerland; 14Institute of Primary Care, University of Zurich, 8006 Zurich, Switzerland; 15National Center of Medicine and Science in Sports (NCMSS), Tunisian Research Laboratory Sports Performance Optimization, Ave Med Ali Akid, El Menzah, Tunis 263, Tunisia

**Keywords:** abdominal muscles, pelvic floor disorders, respiratory function tests, respiratory muscle

## Abstract

Patients with stress urinary incontinence (SUI) may be afraid to increase intra-abdominal pressure to avoid incontinence. This could lead to weak expiratory muscles. The aim of this study was to investigate the association between respiratory muscle strength, physical function, and SUI in patients with SUI. A cross-sectional study was conducted in the Physical Medicine and Functional Rehabilitation Department. Thirty-one incontinent women (IG) and twenty-nine women in a control group (CG) were enrolled in this study. Anthropometric data, respiratory muscle strength (maximal inspiratory pressure; maximal expiratory pressure), SUI (Urogenital Distress Inventory-6; Incontinence Impact Questionnaire-7; Pad test), and physical function (waist circumference; timed-up-and-go test; abdominal muscle strength) were assessed. Body fat, body mass index, body weight, and waist circumference were higher in IG than CG (*p* < 0.01), while postural gait and abdominal muscles were lower (*p* < 0.001). Respiratory muscle strength displayed moderate correlations with SUI severity, especially for maximal expiratory pressure (*p* < 0.01). Maximal expiratory pressure was moderately associated with physical function. Deterioration in respiratory muscle strength is a characteristic of women with SUI. In this population, pelvic floor muscle training may be prescribed to improve continence. By feeling more confident about increasing intra-abdominal pressure, women with SUI would strengthen their expiratory muscles and eventually improve their physical function.

## 1. Introduction

Stress urinary incontinence (SUI) is the most prevalent type of urinary incontinence (UI) and is defined as the complaint of involuntary leakage on effort, exertion, sneezing, or coughing [1]. It is known that SUI primarily results from a pelvic floor muscles (PFM) dysfunction and is associated with an abnormal breathing pattern [2]. These physiological alterations likely affect the synergy between the diaphragm, the abdominal muscles, and the PFM during functional tasks [3].

During tasks requiring a forced expiratory effort (e.g., Valsalva maneuver, coughing, laughing, or sneezing), the combined actions of PFM, abdominal muscles, and the diaphragm help generate an expiratory force, increase intra-abdominal pressure (IAP), and maintain continence [4,5]. The trunk-stabilization muscles (i.e., composed of the abdominal, the pelvic floor, and the respiratory diaphragm muscles) work synchronously to maintain the abdominal lumbopelvic cavity stable by regulating IAP during a variety of daily life activities [6,7,8]. The action of these muscles can alter IAP and thereby influence PFM activity [9]. Recent evidence indicates that modified activation of trunk stabilization muscles could affect urinary continence regulatory mechanisms [10]. In addition, the coactivation of the trunk stabilization muscles could improve PFM-contracting behavior [11,12]. Altogether, the synchronized action between the PFM, the respiratory muscles, and the abdominal wall muscles participate in greater trunk stability that is in turn required to optimally perform tasks of daily life, including locomotor activities [13].

Since abdominal muscles work in synergy with PFM, participate in the respiratory mechanism, and cooperate to support breathing, alteration in one of these muscle groups may have an impact on the function of the other [14]. In light of this, we supposed that SUI may be associated with the changes in respiratory muscle performance.

To the best of the authors’ knowledge, there are few published data on the nature of the association between respiratory muscle strength (RMS) and impaired physical function (PF) in patients with SUI [15]. In addition, the relationship between RMS and SUI is unclear. Therefore, the aims of this comparative study were to (i) compare RMS and PF between SUI and healthy women, (ii) investigate the relationship between RMS and SUI severity, and (iii) evaluate the association of RMS with the PF of women with SUI. It was hypothesized that (i) RMS is compromised in SUI women compared to their healthy counterparts; (ii) RMS, and more particularly expiratory muscle pressure, is negatively correlated with SUI severity; and (iii) impaired PF is associated with both decreased RMS and SUI severity.

## 2. Materials and Methods

### 2.1. Study Design

This study is a subgroup analysis within the ongoing randomized controlled trial registered with the Pan African Clinical Trial Registry (PACTR) (identification number: PACTR202210746628521) to evaluate the impact of inspiratory muscle training on PFM strength and urinary leakage in patients with SUI. A cross-sectional, descriptive study was conducted following the guidelines established by the STROBE statement [16]. Testing occurred between January 2019 and June 2019 at Physical Medicine and Functional Rehabilitation Department, Habib Bourguiba Hospital (Sfax, Tunisia). Anthropometric measurement, RMS, PF, and SUI severity (Urogenital Distress Inventory-6 (UDI-6), Incontinence Impact Questionnaire-7 (IIQ-7), and pad test) were assessed in the morning by the same clinical evaluator. All the participants signed an informed consent form approved by the institutional research ethics committee prior to enrollment in the study. The study was conducted in accordance with the Declaration of Helsinki 1975 and approved by the local ethics committee from the High Institute of Nursing, University of Sfax, Tunisia (Protection Committee Approval Registration code: CPP SUD N° 0355/2021).

### 2.2. Sample Size

A priori sample size was calculated using G∗Power [17] (version 3.1.9.6; University of Kiel, Franz Faul, Germany). Twenty-eight participants were deemed sufficient for an α error probability of 0.05 and power (1-β error probability) of 0.80 and an effect size of 0.8. A final sample size 60 women (31 women with UI (intervention group (IG)) and 29 healthy, age-matched healthy women (control group (CG)) was therefore recruited. Assumption of 56% for the non-inclusion- and exclusion- criteria gave a corrected total sample of 136 women [136 = 60/(1 − 0.56)].

### 2.3. Participants: Intervention and Control Groups

In the IG, women were initially selected from the Physical Medicine and Functional Rehabilitation Department database of the Habib Bourguiba Hospital (Sfax, Tunisia) based on a clinical assessment. The following inclusion criteria were applied: age between 35 and 65 years and any reported unintentional loss of urine during physical activity (walking, running, laughing, sneezing, or coughing) [1] (Figure 1). Participants were excluded if they met any of the following criteria: pelvic prolapse (i.e., a pelvic exam was conducted by a physician to evaluate the presence of prolapse), anal incontinence, urge incontinence, overactive bladder, presence of any concomitant pathology that may affect respiratory and pelvic floor function, history of surgery, recent trauma at the lumbopelvic or the abdominal or the thoracic regions or at the lower limbs, smoking, athletes (i.e., the international physical activity questionnaire short form (IPAQ-SF) was used to verify that tested population was only composed of sedentary women), and nulliparous women. In the CG, which did not report UI, participants were recruited among healthy nurses and from the local community.

### 2.4. Anthropometric Measurements

The height was measured with a Harpenden stadiometer to the last complete 0.1 cm (Seca, Hamburg, Germany). A multi-frequency bioelectrical impedance meter (TBF-410GS, Tanita Co., Tokyo, Japan) was used to measure body weight, body fat, lean mass, and body mass index (BMI) [18]. This method was validated against the reference methods [19]. To obtain an accurate measurement of the patient’s body waist circumference (WC), a stretch-resistant tape that provides a constant tension of 100 g was used by only one trained investigator to minimize measurement errors and variability. In addition, the standard operating procedure was followed when conducting the measurements [18]. The WC was measured at the approximate midpoint between the lower margin of the last palpable rib and the top of the iliac crest [20]. The WC was utilized to determine central obesity beyond the women’s threshold value of 88 cm. Tested individuals remained calm, and measurements were taken at the end of a normal expiration. Each measurement was taken twice, and the average was taken if the two measurements were within 1 cm of each other. The measurements were repeated if the difference between the two measurements was more than 1 cm.

### 2.5. Urogenital Distress Inventory-6 (UDI-6)

The Arabic-validated short form of the UDI-6, which is reliable with an intraclass correlation coefficient (ICC) = 0.98, was used to measure UI severity [21]. Briefly, this self-administered questionnaire comprises six scored items, scaled from 0 to 3, to assess the prevalence, frequency, and severity of urinary leakage [22]. The total score ranges from 0 to 100, with the highest values demonstrating increased severity [23,24]. Items and questions were carefully explained by the interviewer.

### 2.6. Incontinence Impact Questionnaire-7 (IIQ-7)

The Arabic-validated version of the IIQ-7 was used to evaluate the quality of life of women with lower urinary tract symptoms [21]. This questionnaire is reliable (i.e., ICC = 0.98) for self-reported outcome measures [21]. The IIQ-7 level of validation, according to International Consultation on Incontinence (ICI) grades, is A [22]. It consists of seven items that are subdivided into four domains: physical activity (items 1 and 2), travel (items 3 and 4), social activities (item 5), and emotional health (items 6 and 7). The total score ranges from 0 to 100, with the highest values indicative of the worse quality of life [25].

### 2.7. The One-Hour Pad Test (Pad Test)

The pad test was conducted in accordance with International Continence Society (ICS) guidelines [26]. All participants were given pre-weighed pads to wear, and after that, they were told to consume 500 mL of plain water over the course of 15 min. Subsequently, women were required to carry out the typical ICS provocation activities [26]. All participants were asked not to urinate during that hour unless it was absolutely essential [26]. The total amount of urine lost may be calculated by reweighing the absorbent pad. The test outcomes are given in grams. When findings for the pad test surpass 2 g, it is termed positive. Greater losses indicate a more serious UI.

### 2.8. Respiratory Muscle Strength (RMS)

RMS was evaluated while participants were seated at rest according to international guidelines [27]. Maximal inspiratory and expiratory pressures (MIP and MEP, respectively) were measured using a digital mouth pressure meter (MicroRPM, Micro Medical Ltd., Rochester, Kent, UK). This device produces adequate MIP and MEP reliability (i.e., ICC > 0.90) [28]. MIP was measured at the functional residual capacity, while MEP was measured following inspiration to maximum total lung capacity using the technique of Black and Hyatt [29]. Since all participants had no previous experience with these maneuvers, great care was taken to explain the procedures fully. Both inhalation and exhalation were performed as quickly as possible while maintaining maximum effort for at least 1 s. With the use of a nose clip, measurements were repeated until three successive trials with a difference of less than 10% between them were observed [27,30,31]. The highest value was used for further data analysis. Given the lack of Tunisian or North African norms for MIP and MEP [32], the latter was compared to the predictive value of Evans and Whitelaw [33]. Data were expressed as an absolute value (cmH_2_O) and as a percentage of the predicted value (% predicted) adjusted for age. 

### 2.9. Physical Function (PF) 

Functional mobility: The timed-up-and-go (TUG) test is a performance-based measure of functional mobility that was created initially to detect mobility and balance deficits [34]. Participants were required to stand up from a chair with armrests, walk three meters as fast as possible (without running), turn around, return to the chair, and sit down. The test was performed twice, and the best time in seconds to complete this task was used.

Abdominal muscle strength (AMS): The trunk extension and flexion modular (TEF) component attached to the Cybex^®^ Norm II isokinetic dynamometer unit (Lumex, Inc., Ronkonkoma, NY, USA) was used to evaluate AMS at 60°/s and 90°/s. After the participant was positioned and secured in the TEF modular component, the range of motion was calculated. The zero anatomical position was established to be a vertical standing position. Setting the zero anatomical position provided the system with a fixed starting position to calculate the range of motion. There was a maximum range of motion of 50°, with 30° of flexion (−30°) and 20° of extension (+20°) of the trunk [35]. This trunk position represented the range of motion commonly observed during daily life activities [36]. Reductions in hip flexion and extension can be achieved by restricting trunk motion to less than 50°, according to Garcia-Vaquero et al. [37]. Furthermore, the placement of the dynamometer’s axis of rotation at the anterior superior iliac spine level and the use of the pad behind the sacrum and the pelvic strap reduced hip motion during the protocol. Concentric trunk flexion was assessed at 60°/s and 90°/s angular velocities. These angular velocities were selected for measuring mechanical work since they are considered safe and reliable [38]. Before testing, participants performed a standardized warm-up that consisted of cycling an ergometer for 10 min followed by a thorough explanation of the test procedure. To help the participants become acquainted with the equipment and the research protocol, a familiarization trial consisting of one set of five consecutive submaximal concentric trunk flexion and extension repetitions at 60°/s and 90°/s was conducted before the trunk muscles strength assessment. All participants performed three series of three maximum concentric contractions at 60°/s and five series of five maximum concentric contractions at 90°/s, as previously recommended [39]. Because fewer contractions are required for test evaluation at lower velocities (i.e., 60°/s), and more contractions are required for test evaluation at higher velocities (i.e., 90°/s), the difference in the amount of series between both velocities was adopted as recommended [40]. During a 10 s contraction, flexion and extension measurements were taken at 60° and 90° of trunk flexion. There was a 30 s rest period between each angular velocity recording. A one-minute rest period between the two measured velocities was observed. All trials were conducted at the same time of day and were overseen by the same researcher. Participants were allowed to view the Cybex NORM computer monitor, and they were encouraged through a standardized “stronger, faster” verbal command during the test session to exert maximal physical effort. The peak torque (N.m) was recorded. When the peak torque’s coefficient of variation (CV) was higher than 25%, the participants were allowed to rest, and the set was repeated [40,41].

### 2.10. Statistical Analysis 

The Kolmogorov–Smirnov statistic was used to test the normality of the distribution of all variables. Data were presented as the mean and standard deviation (SD) or as the median and interquartile range (25th to 75th percentile). Anthropometric parameters, PF and RMS of IG and CG were compared between the two groups using Student’s *t*-test when the normality of distribution (Kolmogorov–Smirnov test) and the equality of variance (Levene median test) was verified. When these conditions were not obtained, a Mann–Whitney rank-sum test was used instead. Spearman’s rank correlation coefficients “r” were used to determine relationships between RMS, UI, functional mobility, and AMS in patients with UI. Magnitude of “r” values was considered as trivial (r < 0.1), small (0.1 < r < 0.3), moderate (0.3 < r < 0.5), large (0.5 < r < 0.7), very large (0.7 < r < 0.9), nearly perfect (r > 0.9), and perfect (r = 1.0) [42]. For the parametric tests, the effect sizes were given by Cohen’s d (mean 1 - mean 2)/pooled SD). For the non-parametric test, the effect size was calculated as r = $\frac{\left|z\right|}{\sqrt{n1+n2}}$. These effect sizes were interpreted as small: 0.2; medium: 0.5; large > 0.8 [43]. We applied multiple linear regression models to analyze associations between body composition, functional mobility, AMS, and SUI (pad test and UDI-6) with RMS. Unadjusted analyses were performed, and 95% confidence intervals (CI) were calculated. All statistical analysis was carried out using the IBM SPSS^®^ Statistics version 25 (IBM Corp., Armonk, NY, USA). Differences were considered significant for *p* < 0.05.

## 3. Results 

### 3.1. Participant Characteristics

One hundred and thirty-nine women were recruited (110 with SUI and 29 CG). After applying the inclusion and exclusion criteria, the final sample consisted of 60 women (IG (*n* = 31), CG (*n* = 29)) (Figure 1).

Anthropometric data, SUI tests, PF, and RMS for CG (*n* = 29) and IG (*n* = 31) are shown in Table 1. No significant differences were found between the two groups for age and body height (*p* > 0.05); however, body weight, fat mass, BMI, and WC scores were significantly higher for IG than CG (*p* = 0.01; *p* < 0.01; *p* < 0.001, and *p* < 0.001, respectively). PF was deteriorated for IG compared to CG (*p* < 0.001), as evidenced by results for TUG (*p* < 0.001) and AMS at both 60°/s and 90°/s (*p* < 0.001). Furthermore, IG had a lower RMS (i.e., MIP and MEP, *p* < 0.001) compared to CG. Finally, IG showed respiratory muscle weakness when compared with normal predictive values (MIP: *p* < 0.001 and MEP: *p* = 0.001). 

### 3.2. Correlation of RMS (MIP and MEP) with SUI (UDI-6, IIQ-7, and Pad Test), PF (TUG, AMS at 60°/s, and AMS at 90°/s), WC, and BMI

A large correlation was observed with MEP for UDI-6, IIQ-7, TUG, AMS at 60°/s, and AMS at 90°/s (Table 2). A moderate correlation was observed with MEP for the pad test and WC, whereas MIP had a moderate correlation with IIQ-7 and AMS at 60°/s. However, no correlation was found between RMS and BMI. (Table 2).

### 3.3. Correlation of SUI (UDI-6, IIQ-7, and Pad Test) with PF (TUG, AMS at 60°/s, and AMS at 90°/s), WC, and BMI

UDI-6 showed a very large correlation with IIQ-7, pad test, and TUG and a large correlation with WC, BMI, AMS at 60°/s, and AMS at 90°/s (Table 2). IIQ-7 showed a very large correlation with the pad test, TUG, and AMS at 60°/s; a large correlation with AMS at 90°/s; and a moderate correlation with WC (Table 2). The pad test showed a large correlation with WC, TUG, and AMS at 60°/s and a moderate correlation with AMS at 90°/s (Table 2).

### 3.4. Correlation of WC and BMI with PF (TUG, AMS at 60°/s, and AMSat 90°/s)

WC significantly correlated with TUG and AMS at 60°/s (Table 2), whereas BMI had a moderate correlation with TUG.

### 3.5. Associations between PF (TUG, AMS at 60°/s), WC, BMI, Fat Mass, and SUI (UDI-6 and Pad Test) with RMS

The linear regression model results are presented in Table 3. We showed that women having a higher fat mass perform more efficiently in the MEP assessment (β coefficient = 0.463, *p* = 0.034). MEP was significantly affected by BMI. Each point increase in BMI leads to alter MEP performance by −1.135. 

The results showed that an increasing score of UDI-6 is associated with worse MEP. Indeed, for each point increase in the mean overall UDI-6 score, the MEP will decrease by 0.354. In addition, MEP score increased with the increase of AMS (β coefficient = 0.161, *p* = 0.041).

However, our results showed that MIP does not present a significant association with SUI, BMI, WC, TUG, or AMS.

## 4. Discussion

This study showed that respiratory muscle weakness leads to a higher degree of UI severity and a deterioration in functional mobility. In particular, expiratory muscle weakness had a negative effect on SUI and PF. This study provides important novel insight into the relationships among RMS, UDI-6, IIQ-7, PF, AMS, WC, and BMI of patients with SUI. These findings are confirmed by the functional connection between the diaphragm, abdominal muscles, and PFM [9], which participate in postural control and breathing [7,44].

The present results support the hypothesis that expiratory muscle contractions during sneezing or coughing generate a rapid increase in IAP, which pulls up the diaphragm and elevates IAP, resulting in high expiratory flow rates [7,45]. Therefore, SUI emerges when powerful contractions of the PFM fail or cannot sustain the increasing pressure resulting from expiratory muscle activity [15]. Consequently, if the expiratory muscle strength is compromised because of its synergic activity, the actions of PFM may also be negatively impacted, potentially leading to SUI. Additionally, the relationship between abdominal muscle and SUI as well as the relationship between SUI and functional mobility could be explained by expiratory muscular strength. The present study produced results that corroborate the findings by Aguilar-Zafra et al. (2022) [15], who found that expiratory muscular weakness had a similar decrease in SUI function.

Our findings identified that expiratory muscle strength was inversely correlated with the WC. In other words, linear regression model results indicate that MEP is associated with BMI. Previous research has linked WC to SUI in the elderly [46,47], but the relationship between BMI and SUI is inconsistent [48]. This could be explained by the fact that a high BMI, which could be due to muscle weight or edema, does not represent abdominal adiposity. However, WC can negatively affect the mechanism of continence through increasing IAP and exerting direct pressure on the pelvic and urethral structures [49,50]. Although WC and BMI are correlated, in practice, measuring and interpreting WC is easier than measuring BMI. As a result, we believe WC is a more accurate clinical predictor of SUI than BMI.

Another key finding was that RMS was associated with AMS, especially for expiratory muscles. This observation supports the role of the contraction of the abdominal muscles during forced expiration maneuvers, which contribute greatly to making expiration more rapid and more efficient. Their actions result in pulling down the ribcage and facilitating the upward movement of the diaphragm [7]. During efforts that increase IAP (e.g., coughing, sneezing, jumping), the PFM is engaged to support the position of the bladder neck and assist in maintaining continence. Several studies support the association between trunk flexor strength and the PFM and suggest that contractions of the PFM are related to abdominal muscle activity [9,51]. From this perspective, it is compelling that trunk or AMS and coordination can generate an appropriate contraction of the PFM, which directly prevents SUI episodes [52,53].

The present study reveals a positive correlation between SUI and expiratory muscle strength in incontinent women for the first time. By measuring expiratory muscle strength, clinicians can detect the deterioration of PF in patients with SUI. Thus, it is recommended to integrate RMS measurements in patients with SUI to prevent some comorbidities [54] (i.e., urinary tract infections, constipation, chronic obstructive pulmonary disease, depression), stop SUI’s negative effect on lifestyle, and reduce the impairment of quality of life. In this vein, some researchers have found that PFM exercise improves pulmonary function and posture [44,55,56], especially when paired with abdominal muscle training [57]. The ability to perform activities of daily living without assistance is commonly used to assess functional capacity. Incontinent women had impairment in functional capacity, and the state of the expiratory muscles can be suggested as an appropriate variable for estimating functional capacity.

The poor performance on TUG tests may be a risk factor for SUI. The objective of the TUG test is to evaluate the dynamic balance between sitting, standing, and walking. The reduction in mobility associated with the functional decline of TUG may be the cause of the development of SUI because of uncoordinated PFM, inability to rush to the toilet in time, or other multifactorial factors [53,58]. 

In addition, as determined by TUG performance, functional mobility impairment was statistically associated with weaker AMS, abdominal obesity, and poorer expiratory muscle strength but not with the inspiratory muscles. These findings, however, could also be attributed to expiratory muscle strength directly interfering with the trunk stabilizing system [44] given that the trunk stabilizing system is associated with balance and walking ability [59]. In support, functional capacity is negatively affected by weakness in the abdominal muscles [59] and a lower balance capacity [60] compared to continent women. As a result, it is reasonable to expect that when patients have difficulty performing activities requiring a significant contribution from abdominal muscles, deficiencies in their ability to perform these activities will be a sign of expiratory muscle weakness. This weakness is associated with alteration in the activity of the PFM, resulting in a shift in the pelvic position, which can impair breathing and posture [61].

Some strengths of the present study should be highlighted. First, we used validated questionnaires to assess the SUI severity. Second, this is the first study that investigated the association between SUI and PF and RMS in North African women with SUI. Future studies with more heterogeneous samples of women (i.e., athletes, pregnant) are warranted. 

This study presents some limitations. First, our study has a cross-sectional design with a relatively small number of recruited participants. Studies with a larger sample size to confirm our findings are necessary. Second, SUI evaluated by the one-hour pad test as suggested by ICS, which is secondary to fluid intake, is sadly unreliable, as patients will have different degrees of dehydration. Therefore, the pad testing needs to be either 48 h or should be with a fixed volume instilled in the bladder. Third, it was not possible to isolate the participation of other muscles that have helped with trunk flexion during the isokinetic dynamometer assessment. Even though the abdominal group is the primary trunk flexor, other muscles, such as the iliopsoas, contribute to trunk flexion and may have influenced strength measurements and, by extension, correlational analysis. Fourth, it would be interesting to determine if respiratory muscle weakness influences the ability to walk short distances by implementing functional tests (e.g., 6 min walk test). Finally, we did not determine the parity of the participants [62]. Parity apparently influences some lung function data, including MIP and MEP [63,64].

## 5. Conclusions

The study showed that incontinent women have lower RMS, mainly within the expiratory muscles, and deteriorated PF (functional ability, AMS, abdominal obesity) compared to healthy individuals. Furthermore, the patients’ urinary and PF were moderately to significantly affected by expiratory muscle strength. Since SUI may decrease PF and RMS, it is highly recommended to encourage women to train their PFM. PFM training could effectively improve RMS and AMS in patients with SUI. By feeling more confident about increasing IAP, women with SUI would strengthen their expiratory muscles and improve PF.

## Figures and Tables

**Figure 1 jcm-11-07344-f001:**
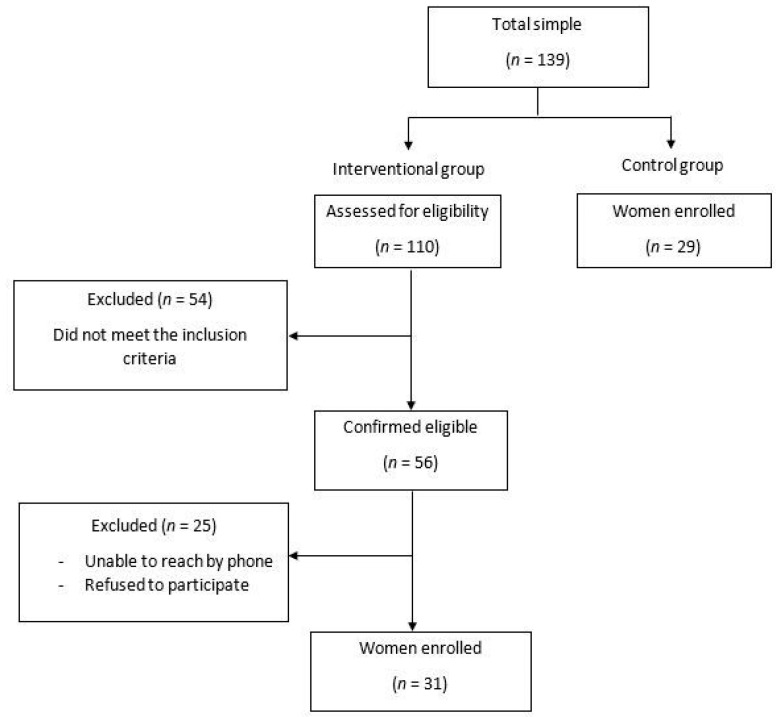
Study design.

**Table 1 jcm-11-07344-t001:** Characteristics of the participants.

	Incontinence Group (*n* = 31)	Control Group (*n* = 29)	*p*-Value	Effect Size
Age (years), mean (SD)	53.2 (7.3)	51.2 (6.1)	0.25 ^a^	0.15 ^z^
Body height (m), mean (SD)	1.6 (0.1)	1.6 (0.06)	0.09 ^a^	0 ^z^
Body weight (kg), median (IQR)	72.6 (67.1 to 80.7)	69.2 (64.to 72.6)	0.01 ^b^	0.30 ^z^
Fat mass (kg), mean (SD)	28.1 (3.2)	25.2 (4.9)	0.005	0.35^z^
Body mass index (kg/m^2^), median (IQR)	27.4 (25.8 to 30.4)	24.8 (24.01 to 26.3)	<0.001 ^b^	0.50 ^z^
Waist circumference (cm), median (IQR)	93 (90 to 100)	84.0 (81 to 89.5)	0.001 ^b^	0.67 ^Q^
Urinary incontinence				
UDI-6, median (IQR)	42 (37.5 to 54)	4.2 (0 to 8.3)	0.001 ^b^	0.86 ^X^
IIQ-7, median (IQR)	41.6 (29.2 to 50)	0.0(0 to 4.2)	<0.001 ^b^	0.87 ^X^
Pad test, median (IQR)	13.0 (8 to 21)	0.2 (0.09 to 0.6)	<0.001 ^b^	0.85 ^X^
Physical function				
TUG (s), median (IQR)	8.9 (7.6 to 9.7)	6.8 (6.4 to 7.1))	<0.001 ^b^	0.72 ^X^
AMS at 60°/s (N.m), median (IQR)	102.0 (72 to 113)	120.0(110.5 to 128))	<0.001 ^b^	0.62 ^z^
AMS at 90°/s (N.m) median (IQR)	90.0 (60 to 102)	106.0 (97.5 to 117)	<0.001 ^b^	0.50 ^z^
RMS (cmH_2_O)				
MIP, mean (SD)	67.2 (5.5)	80.1 (4.8)	<0.001 ^a^	0.78 ^X^
MEP, median (IQR)	79.0 (73 to 86)	89.0 (81.5 to 92.5)	<0.001 ^b^	0.50 ^z^
RMS (% predicted value)				
MIP, median (IQR)	87.5 (81.9 to 96.5)	103.4 (96.8 to 104.8)	<0.001 ^b^	0.73 ^X^
MEP, mean (SD)	93.6 (9.4)	101.1 (5.9)	0.001 ^a^	0.43 ^z^
RMS	Incontinence group	Predicted value	*p*-value	
MIP vs. MIP_ref_, mean (SD)	67.2 (5.5)	75.5 (4.5)	<0.001 ^c^	0.63 ^z^
MEP vs. MEP_ref_, median (IQR)	79 (73 to 86)	83.7 (80 to 89.7)	0.001 ^d^	0.31 ^z^

Values expressed as mean (standard deviation). AMS, abdominal muscle strength; IIQ-7, Incontinence Impact Questionnaire—Short Form; MEP, maximal expiratory pressure, MEP_ref_, references values of MEP; MIP, maximal inspiratory pressure; MIP_ref_, references values of MIP; RMS, respiratory muscle strength; TUG, timed-up-and-go test; UDI-6, Urogenital Distress Inventory—Short Form. ^a^ Independent *t*-test; ^b^ U-Mann–Whitney; ^c^ paired-sample test; ^d^ the Wilcoxon signed-rank test. Effect size was considered as ^z^ small (effect sizes ≤ 0.3); ^Q^ medium (0.3 < effect size ≤ 0.5), or ^X^ large (effect size > 0.5).

**Table 2 jcm-11-07344-t002:** Spearman correlation coefficient “r” analysis for relationships between respiratory muscle strength (RMS), stress urinary incontinence (SUI), waist circumference (WC), body mass index (BMI), and physical function (PF) in incontinent women (*n* = 31).

		RMS		SUI				WC	BMI	PF		
		MIP	MEP	UDI-6	IIQ-7	Pad Test				Functional Mobility (TUG)	AMS at 60°/s	AMS at 90°/s
RMS	MIP	-	0.51 **^b^	−0.28	−0.36 *^a^	−0.09		−0.27	−0.13	−0.32	0.38 *^a^	0.29
	MEP	0.51 **^b^	-	−0.64 **^b^	−0.61 **^b^	−0.45 *^a^		−0.35 *^a^	−0.28	−0.65 **^b^	0.69 **^b^	0.62 *^b^
SUI	UDI-6	−0.28	−0.64 **^b^	-	0.78 **^c^	0.80 **^c^		0.62 **^b^	0.52 **^b^	0.73 **^c^	−0.68 **^b^	−0.53 **^b^
	IIQ-7	−0.36 *^a^	−0.61 **^b^	0.78 **^c^	-	0.74 **^c^		0.47 **^a^	0.29	0.74 **^c^	−0.73 **^c^	−0.58 **^b^
	Pad test	−0.09	−0.45 *^a^	0.80 **^c^	0.74 **^c^	-		0.57 **^b^	0.30	0.67 **^b^	−0.63 **^b^	−0.46 **^a^
WC		−0.27	−0.35 *^a^	0.62 **^b^	0.47 **^a^	0.57 **^b^		-	0.74 **^c^	0.55 **^b^	−0.42 *^a^	−0.26
BMI		−0.13	−0.28		0.52 **^b^	0.29	0.30	0.74 **^c^		0.40 * ^a^	−0.17	0.004
PF	Functional mobility (TUG)	−0.32	−0.65 **^b^	0.73 **^c^	0.74 **^c^	0.67 **^b^		−0.55 **^b^	0.40 *^a^	-	−0.80 **^c^	−0.72 **^c^
	AMS at 60°/s	0.38 *^a^	0.69 **^b^	−0.68 **^b^	−0.73 **^c^	−0.63 **^b^		−0.42 *^a^	−017	−0.80 **^c^	-	0.88 **^c^
	AMS at 90°/s	0.29	0.62 *^b^	−0.53 **^b^	−0.58 **^b^	−0.46 **^a^		−0.26	0.004	−0.72 **^c^	0.88 **^c^	-

AMS, abdominal muscle strength; IIQ-7, Incontinence Impact Questionnaire—Short Form; MEP, maximal expiratory pressure; MIP, maximal inspiratory pressure; SUI, stress urinary incontinence; TUG, timed-up-and-go test; UDI-6, Urogenital Distress Inventory—Short Form. “r” was considered as ^a^ moderate (0.3 < r < 0.5), ^b^ large (0.5 < r < 0.7), and ^c^ very large (0.7 < r < 0.9). * *p* < 0.05. ** *p* < 0.01.

**Table 3 jcm-11-07344-t003:** Regression analysis summary for factors impacting on MEP and MIP for women with SUI.

	Variables	Unstandardized B	95% CI	*p*-Value
MEP	Fat mass	0.463	0.038–0.889	0.034 ^a^
	BMI	−1.135	−2.212–−0.057	0.04 ^a^
	WC	0.277	−0.166–0.720	0.209 ^a^
	Pad test	0.161	−0.368–0.664	0.559 ^a^
	UDI-6	−0.354	−0.691–−0.016	0.041 ^a^
	TUG	−0.153	−2.903–2.597	0.909 ^a^
	AMS at 60°/s	0.161	0.007–0.315	0.041 ^a^
MIP	Fat mass	0.414	−0.023–0.850	0.062 ^a^
	BMI	−0.834	−1.938–0.271	0.132 ^a^
	WC	0.015	−0.439–0.469	0.946 ^a^
	Pad test	0.362	−0.167–0.891	0.170 ^a^
	UDI-6	−0.222	−0.568–0.124	0.197 ^a^
	TUG	0.309	−2.510–3.128	0.823 ^a^
	AMS at 60°/s	0.106	−0.052–0.263	0.180 ^a^

AMS, abdominal muscle strength; BMI, body mass index; MEP, maximal expiratory strength; MIP, maximal inspiratory strength; SUI, stress urinary incontinence; TUG, timed-up-and-go test; UDI-6, Urogenital Distress Inventory—Short Form; WC, waist circumference. ^a^
*p*-values by linear regression.

## Data Availability

Data available on request due to privacy restrictions.

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
