# Peer review of "Impact of Urinary Incontinence on Physical Function and Respiratory Muscle Strength in Incontinent Women: A Comparative Study between Urinary Incontinent and Apparently Healthy Women"

_jcm, 2022, doi:10.3390/jcm11247344_

Round 1
Reviewer 1 Report
This is a well-written and interesting paper. I would still like to make a couple of comments.
1: I am not sure why you have not tried to make two more homogeneous groups? Do you think it would have been interesting to take into account similar values for waist circumference, body mass index or fat mass?
2: References to books should include the pages and their references 44 and 47 do not reflect them.
Author Response
TO REVIEWER #1
Dear Reviewer,
Thank you for your comments. Please find below the responses to your questions/suggestions. Sincerely yours.
Comment 1: I am not sure why you have not tried to make two more homogeneous groups? Do you think it would have been interesting to take into account similar values for waist circumference, body mass index or fat mass?
Response: Thank you for your comments.
Actually, the IG was composed of non-active women or sedentary lifestyle is associated with a high risk of urinary incontinence and being active alleviated this risk. In addition, being non-active favors obesity and high BMI. Furthermore, it has been reported that women with urinary incontinence will restrict their activities and may stop practicing sports which will complicate the situation and provoke weight gain [1]. For that reason, we can explain why incontinent females have high values of waist circumference, body mass index and fat mass. But it looks interesting to compare expiratory muscle strength in incontinent and continent women whose body composition seems homogeneous.
Comment 2: References to books should include the pages and their references 44 and 47 do not reflect them
Response: Thank you for your comments.
We verified the missing information in references and we replaced references with others indicating the number of pages
Reference 44 was replaced by this reference: “Sole G, Hamrén J, Milosavljevic S, et al. Test-retest reliability of isokinetic knee extension and flexion. Archives of physical medicine and rehabilitation. 2007;88(5):626-631”
Reference 47 was replaced by this reference: “Cohen J. Statistical power analysis for the behavioral sciences. Hoboken. NJ: Taylor and Francis. 2013:23-26”.
Reviewer 2 Report
This is a case-control study comparing respiratory data between incontinent vs continent women using a subgroup analysis from a RCT. The study findings yield significant associations between respiratory muscle strength metrics and incontinence severity. From a urological perspective, the study requires revisions if the data is potentially available. The term urinary incontinence is broad and can encompass not only stress urinary incontinence, but also urgency incontinence, unaware incontinence, and overflow incontinence. While the authors do cite a study [6] from NAU, which does not distinguish between these types of incontinence in their experimental study, a retrospective analysis should make an attempt to distinguish this. This is because the anatomical mechanism that they hypothesize during valsalva-induced incontinence is primarily STRESS urinary incontinence or possibly stress-induced urgency. The authors exclude patients with multiple conditions but choose an age group that could easily have overactive bladder with urinary urgency incontinence. Moreover, the incontinence group has significantly higher metrics for obesity, which should be considered a confounding variable that would need a regression model added to the analysis. They do justify this in the DISCUSSION by saying the data is not established on the relationship between respiratory metrics and obesity, but the relationship should still be explored in this study.
INTRODUCTION
Would change urinary incontinence to STRESS URINARY INCONTINENCE for all purposes of this study. The mechanism of urinary urgency incontinence in overactive bladder is very different from the one discussed in this introduction. Also overflow urinary incontinence is likely due to mechanical factors such as BPH, or neurological factors, such as spinal cord injury.
STUDY DESIGN
Can the authors distinguish what type of urinary incontinence the patients have?
Why were nulliparous women excluded? Did any of the women have a diagnosis of Overactive or Neurogenic Bladder and could they be excluded?
UDI-6 - were the scores of individual questions available for subanalysis? It has been established that items 3 & 4 pertain more to the domain of STRESS URINARY INCONTINENCE. Item 5 pertains to OVERFLOW INCONTINENCE or incomplete bladder emptying. Items 1 & 2 are clearly pertaining to OAB or Urinary Urgency Incontinence. A sub-analysis for only the scores of 3+4 could be meaningful.
Statistics
Waist circumference and/or BMI needs to be evaluated as a major confounding variable for these findings. They are significantly higher in the incontinence group and this warrants the best method to elucidate this, which would be a regression model with BMI and waist circumference as likely variables.
Overall this study is well organized and the findings are presented in a thoughtful and well-written manner.
Author Response
TO REVIEWER #2
Dear Reviewer,
Thank you for your comments. Please find below the responses to your questions/suggestions. Sincerely yours. Comment 1: This is a case-control study comparing respiratory data between incontinent vs continent women using a subgroup analysis from a RCT. The study findings yield significant associations between respiratory muscle strength metrics and incontinence severity. From a urological perspective, the study requires revisions if the data is potentially available. The term urinary incontinence is broad and can encompass not only stress urinary incontinence, but also urgency incontinence, unaware incontinence, and overflow incontinence. While the authors do cite a study [6] from NAU, which does not distinguish between these types of incontinence in their experimental study, a retrospective analysis should make an attempt to distinguish this. This is because the anatomical mechanism that they hypothesize during valsalva-induced incontinence is primarily STRESS urinary incontinence or possibly stress-induced urgency. The authors exclude patients with multiple conditions but choose an age group that could easily have overactive bladder with urinary urgency incontinence. Moreover, the incontinence group has significantly higher metrics for obesity, which should be considered a confounding variable that would need a regression model added to the analysis. They do justify this in the DISCUSSION by saying the data is not established on the relationship between respiratory metrics and obesity, but the relationship should still be explored in this study.
Response: Thank you for this important comment. Starting with STRESS URINARY INCONTINENCE (SUI). Our study is composed of women who suffer from SUI. It is important to indicate that among the inclusion criteria, women who suffer from SUI (a self-reported UI was assessed using the following question: in the last month did you experience leaking urine related to urgency, or did you experience urine leakage related to physical activity (walking, running, laughing, sneezing, or coughing) those who responded as heaving urge leakage where excluded from the study and women who responded as having urine leakage related to physical activity were classified as having stress UI). Therefore, having urge incontinence or overactive bladder were classified as exclusion criteria. Furthermore, the UDI-6 questionnaire approved this classification. We added this to the study:
Concerning obesity, we explained this point previously. In addition, a regression model was added to the analysis to verify expiratory muscle alteration in incontinent females. Besides we add in the correlation table the correlation of the BMI with other variables.
Comment 2: INTRODUCTION: Would change urinary incontinence to STRESS URINARY INCONTINENCE for all purposes of this study. The mechanism of urinary urgency incontinence in overactive bladder is very different from the one discussed in this introduction. Also overflow urinary incontinence is likely due to mechanical factors such as BPH, or neurological factors, such as spinal cord injury.
Response: We agree with this comment. Consequently, we modified UI with stress urinary incontinence as we explained in the heading response.
Line 60: we replace “Urinary incontinence (UI), “the complaint of any involuntary leakage of urine”[2] is associated with various alterations and limitations to people's lives. UI has a negative impact on the physical [3], emotional [4] and social health [5], and well-being [6]. » by “ Stress urinary incontinence (SUI) is the most prevalent type of UI and is defined as the complaint of involuntary leakage on effort or exertion, or on sneezing, or coughing [7].”
Line 64: we replace UI with SUI: “It is known that SUI primarily results from a pelvic floor muscles (PFM) dysfunction and is associated with an abnormal breathing pattern [8].
Comment 3: STUDY DESIGN: Can the authors distinguish what type of urinary incontinence the patients have?
Response: We have added this detail in the section of “study design”
Comment 4: Why were nulliparous women excluded? Did any of the women have a diagnosis of Overactive or Neurogenic Bladder and could they be excluded?
Response: Based on the study of Afshari et al 2017[9], Nulliparous women had a higher pelvic floor muscle strength compared to multiparous women regardless of the nature of delivery. Therefore, we exclude nulliparous women to verify that all our simple is composed of only multiparous women. Therefore, we find it necessary to put nulliparity as an excluded criterion.
No women have a diagnosis of overactive or neurogenic bladder. As a reason of fact having overactive or neurogenic bladder were classified as an exclusion criterion
Comment 5: UDI-6 - were the scores of individual questions available for subanalysis? It has been established that items 3 & 4 pertain more to the domain of STRESS URINARY INCONTINENCE. Item 5 pertains to OVERFLOW INCONTINENCE or incomplete bladder emptying. Items 1 & 2 are clearly pertaining to OAB or Urinary Urgency Incontinence. A sub-analysis for only the scores of 3+4 could be meaningful.
Response: Certainly, the Urogenital Distress Inventory Short Form (UDI-6) assesses symptom distress in women suffering from urinary incontinence[10]. It is important to note that the UDI-6 is a self-reported questionnaire that it may have some bias; however, the results of our simple indicated that participants had major urinary incontinence related to physical activity. Thus, we added this interesting point in the “Participants:” section.
Comment 6: Statistics: Waist circumference and/or BMI needs to be evaluated as a major confounding variable for these findings. They are significantly higher in the incontinence group and this warrants the best method to elucidate this, which would be a regression model with BMI and waist circumference as likely variables.
Response: a regression model was added to the analysis to verify expiratory muscle alteration in incontinent female
3.5. Associations between physical function (TUG, abdominal muscle strength at 60°/s), WC, BMI, fat mass, UI (UDI-6, and Pad test) with RMS:
Table 2. Regression analysis summary for factors impacting on MEP and MIP for women with UI,
|
Variables |
Unstandardized B |
95% CI |
P value |
MEP |
Fat mass |
0.463 |
0.038 - 0.889 |
0.034a |
BMI |
-1.135 |
-2.212 - -0.057 |
0.04a |
|
WC |
0.277 |
-0.166 - 0.720 |
0.209a |
|
Pad test |
0.161 |
-0.368 - 0.664 |
0.559a |
|
UDI-6 |
-0.354 |
-0.691 - -0.016 |
0.041a |
|
TUG |
-0.153 |
-2.903 - 2.597 |
0.909a |
|
AMS at 60°/s |
0.161 |
0.007 - 0.315 |
0.041a |
|
MIP |
Fat mass |
0.414 |
-0.023 – 0.850 |
0.062a |
BMI |
-0.834 |
-1.938 – 0.271 |
0.132a |
|
WC |
0.015 |
-0.439 – 0.469 |
0.946a |
|
Pad test |
0.362 |
-0.167 – 0.891 |
0.170a |
|
UDI-6 |
-0.222 |
-0.568 – 0.124 |
0.197a |
|
TUG |
0.309 |
-2.510 – 3.128 |
0.823a |
|
AMS at 60°/s |
0.106 |
-0.052 – 0.263 |
0.180a |
MEP: maximal expiratory strength, MIP: maximal inspiratory strength; AMS: Abdominal muscle strength; TUG: Timed Up & Go test; UDI-6: Urogenital distress inventory short form; WC: Waist Circumference. BMI; body mass index; ap values by Linear regression.
The linear regression model results are presented in table 2. We showed that women having a higher fat mass perform more efficiently in MEP assessment (β coefficient = 0. 463, p = 0.034). MEP was significantly affected by BMI. Each point increase in BMI leads to alter MEP performance by -1.135.
The results showed that an increasing score of UDI-6 is associated with worse MEP. Indeed, for each point increase in the mean overall UDI-6 score, the MEP will decrease by 0.354. In addition, MEP score increased with the increase of AMS at 60°/s (β coefficient = 0. 161, p = 0.041)
However, our results showed that MIP does not present significant association with UI, BMI, WC, TUG, or AMS
References:
- Sazonova, N.A., et al., [Urinary incontinence in women and its impact on quality of life]. Urologiia, 2022(2): p. 136-139.
- Abrams, P., et al., The standardisation of terminology of lower urinary tract function: report from the Standardisation Sub-committee of the International Continence Society. Neurourol Urodyn, 2002. 21(2): p. 167-78.
- Brown, W.J. and Y.D. Miller, Too wet to exercise? Leaking urine as a barrier to physical activity in women. Journal of Science and Medicine in Sport, 2001. 4(4): p. 373-378.
- Irwin, D.E., et al., Impact of overactive bladder symptoms on employment, social interactions and emotional well-being in six European countries. BJU International, 2006. 97(1): p. 96-100.
- Fultz, N.H. and A.R. Herzog, Self-Reported Social and Emotional Impact of Urinary Incontinence. Journal of the American Geriatrics Society, 2001. 49(7): p. 892-899.
- Heidrich, S.M. and T.J. Wells, EFFECTS OF Urinary Incontinence: Psychological Well-Being and Distress in Older Community-Dwelling Women. Journal of Gerontological Nursing, 2004. 30(5): p. 47-54.
- Haylen, B.T., et al., An International Urogynecological Association (IUGA)/International Continence Society (ICS) joint report on the terminology for female pelvic floor dysfunction. Int Urogynecol J, 2010. 21(1): p. 5-26.
- Thompson, J.A., et al., Altered muscle activation patterns in symptomatic women during pelvic floor muscle contraction and Valsalva manouevre. Neurourol Urodyn, 2006. 25(3): p. 268-276.
- Afshari, P., et al., Comparison of pelvic floor muscle strength in nulliparous women and those with normal vaginal delivery and cesarean section. International Urogynecology Journal, 2017. 28(8): p. 1171-1175.
- Skorupska, K., et al., Identification of the Urogenital Distress Inventory-6 and the Incontinence Impact Questionnaire-7 cutoff scores in urinary incontinent women. Health and Quality of Life Outcomes, 2021. 19(1): p. 1-6.
Round 2
Reviewer 2 Report
Abstract:
Need to define acronym UI in 3rd sentence
or can change all remaining UI --> SUI
Author Response
TO REVIEWER #2
Dear Reviewer,
Thank you for your comments. Please find below the responses to your questions/suggestions. Sincerely yours.
Comment 1: Abstract: Need to define acronym UI in 3rd sentence or can change all remaining UI --> SUI
Response: Thank you for your comments.
The following change was done: We have changed the acronym UI in 3rd sentence in the abstract by SUI.
“The aim of this study was to investigate the association between respiratory muscle strength (RMS), physical function (PF) and SUI in patients with SUI.”
We have checked the reference list and citation style as recommended by the ACS style guide.
As well, we have verified the manuscript again and made the appropriate correction.